

# A systematic comparison of eight new plastome sequences from *Ipomoea* L

Jianying Sun[1,2,*], Xiaofeng Dong[1,2,*], Qinghe Cao[3], Tao Xu[1,2], Mingku Zhu[1,2], Jian Sun[1,2], Tingting Dong[1,2], Daifu Ma[3], Yonghua Han[1,2] and Zongyun Li[1,2]

[1] Institute of Integrative Plant Biology, School of Life Sciences, Jiangsu Normal University, Xuzhou, China
[2] Jiangsu Key Laboratory of Phylogenomics and Comparative Genomics, Jiangsu Normal University, Xuzhou, China
[3] Jiangsu Xuhuai Regional Xuzhou Institute of Agricultural Sciences, Chinese Academy of Agricultural Sciences, Xuzhou, China
* These authors contributed equally to this work.

Corresponding authors
Yonghua Han,
hanyonghua@jsnu.edu.cn
Zongyun Li, zongyunli@jsnu.edu.cn

## ABSTRACT

**Background:** *Ipomoea* is the largest genus in the family Convolvulaceae. The species in this genus have been widely used in many fields, such as agriculture, nutrition, and medicine. With the development of next-generation sequencing, more than 50 chloroplast genomes of *Ipomoea* species have been sequenced. However, the repeats and divergence regions in *Ipomoea* have not been well investigated. In the present study, we sequenced and assembled eight chloroplast genomes from sweet potato's close wild relatives. By combining these with 32 published chloroplast genomes, we conducted a detailed comparative analysis of a broad range of *Ipomoea* species.
**Methods:** Eight chloroplast genomes were assembled using short DNA sequences generated by next-generation sequencing technology. By combining these chloroplast genomes with 32 other published *Ipomoea* chloroplast genomes downloaded from GenBank and the Oxford Research Archive, we conducted a comparative analysis of the repeat sequences and divergence regions across the *Ipomoea* genus. In addition, separate analyses of the Batatas group and Quamoclit group were also performed.
**Results:** The eight newly sequenced chloroplast genomes ranged from 161,225 to 161,721 bp in length and displayed the typical circular quadripartite structure, consisting of a pair of inverted repeat (IR) regions (30,798–30,910 bp each) separated by a large single copy (LSC) region (87,575–88,004 bp) and a small single copy (SSC) region (12,018–12,051 bp). The average guanine-cytosine (GC) content was approximately 40.5% in the IR region, 36.1% in the LSC region, 32.2% in the SSC regions, and 37.5% in complete sequence for all the generated plastomes. The eight chloroplast genome sequences from this study included 80 protein-coding genes, four rRNAs (rrn23, rrn16, rrn5, and rrn4.5), and 37 tRNAs. The boundaries of single copy regions and IR regions were highly conserved in the eight chloroplast genomes. In *Ipomoea*, 57–89 pairs of repetitive sequences and 39–64 simple sequence repeats were found. By conducting a sliding window analysis, we found six relatively high variable regions (*ndhA* intron, *ndhH-ndhF*, *ndhF-rpl32*, *rpl32-trnL*, *rps16-trnQ*, and *ndhF*) in the *Ipomoea* genus, eight (*trnG*, *rpl32-trnL*, *ndhA* intron, *ndhF-rpl32*, *ndhH-ndhF*, *ccsA-ndhD*, *trnG-trnR*, and *pasA-ycf3*) in the Batatas group, and eight (*ndhA* intron, *petN-psbM*, *rpl32-trnL*, *trnG-trnR*, *trnK-rps16*, *ndhC-trnV*, *rps16-trnQ*,

and *trnG*) in the Quamoclit group. Our maximum-likelihood tree based on whole chloroplast genomes confirmed the phylogenetic topology reported in previous studies. **Conclusions:** The chloroplast genome sequence and structure were highly conserved in the eight newly-sequenced *Ipomoea* species. Our comparative analysis included a broad range of *Ipomoea* chloroplast genomes, providing valuable information for *Ipomoea* species identification and enhancing the understanding of *Ipomoea* genetic resources.

# INTRODUCTION

*Ipomoea* belongs to the family Convolvulaceae comprising 600–700 species and is distributed throughout the tropical and subtropical regions (*Austin & Huáman, 1996*). *Ipomoea* mainly contains three subgenera: *Eriospermum*, *Quamoclit*, and *Ipomoea*. The species in this genus have been used in many fields, such as nutrition, medicine, rituals, and agriculture (*Meira et al., 2012*). Sweet potato, *Ipomoea batatas* (L.) Lam, is one of the most important food crops in the world and is grown in more than 100 countries (*Food and Agriculture Organization of the United Nations, 2017*).

Sweet potato is in *Ipomoea* series *Batatas*. After few corrections, more than 16 species were included in *I.* ser. *Batatas* (*Austin, 1987*; *McDonald & Austin, 1990*; *Muñoz-Rodríguez et al., 2018*). The phylogenetic relationships in *I.* ser. *Batatas* and the evolutionary origin of *I. batatas* have been a research focus for many years. Molecular markers, such as restriction fragment length polymorphisms (RFLPs) of genomic DNA, random amplified polymorphic DNAs (RAPDs), inter-simple sequence repeats (ISSRs), and some gene sequences have been used to investigate phylogenetic relationships of *I. batatas* and its wild relatives (*Gao et al., 2011*; *Rajapakse et al., 2004*; *Huang & Sun, 2000*; *Jarret & Austin, 1994*; *Jarret, Gawel & Whittemore, 1992*). In *Ipomoea*, various studies including morphology of pollen, flower, and seed; systemic characteristics, and floral anthocyanin regulators have been conducted (*Jayeola & Oladunjoye, 2012*; *Das, 2011*; *Streisfeld & Rausher, 2007*).

With the development of next-generation sequencing, large amounts of genomic data have been obtained to solve the taxonomic relationships in plants. Of these, chloroplast genome sequences are one of the most important sources. The chloroplast, the photosynthetic organelle in plants, has its own genome. The chloroplast genome of *Nicotiana tabacum* was the first chloroplast genome to be sequenced in 1986 (*Shinozaki et al., 1986*). The lengths of land-dwelling plant chloroplast genomes are generally 120–165 kb, the chloroplast DNA inheritance is mostly maternal and the genome is highly conserved in gene content and structure (*Harris & Ingram, 1991*; *Raubeson & Jansen, 2005*). Chloroplast genomes carry a large amount of valuable phylogenetic information and have been widely used to study the evolutionary relationships at almost any taxonomic level in plants (*Tong, Kim & Park, 2016*; *Zhang et al., 2016*; *Carbonell-Caballero et al., 2015*;

**Table 1 The information of the eight species.**

| Species | Plant ID | Maintained by | Origin | GenBank accession number |
|---|---|---|---|---|
| *I. trifida* | CIP 460377 | CIP | Nicaragua | MH173261 |
| *I. triloba* | NCNSP0323 | North Carolina State University | USA, North Carl | MH173262 |
| *I. lacunosa* | Grif 6172 | NPGS(S9) | USA, South Carl | MH173257 |
| *I. × leucantha* | PI 540733 | NPGS(S9) | Colombia, Cesar | MH173263 |
| *I. cynanchifolia* | PI 549093 | NPGS(S9) | Peru | MH173253 |
| *I. splendor-sylvae* | PI 561557 | NPGS(S9) | Mexico | MH173259 |
| *I. cordatotriloba* | PI 518495 | NPGS(S9) | Mexico, Tabasco | MH173254 |
| *I. tabascana* | PI 518479 | NPGS(S9) | Mexico, Tabasco | MH173260 |

Note:
CIP, International Potato Center; NPGS, US National Plant Germplasm System; S9, Plant Genetic Resources Conservation Unit, Griffin, GA.

*Jansen et al., 2007*). In *Ipomoea*, *Eserman et al. (2014)* utilized chloroplast genomes to properly resolve the phylogenetic relationships. *Muñoz-Rodríguez et al. (2018)* explained the origin of sweet potato by chloroplast genome sequence combined with nuclear regions. Although many chloroplast genomes of the *Ipomoea* genus have been obtained and have undergone phylogeny analysis, the divergence regions and variations of repeats have not been studied in the *Ipomoea* genus.

In this study, we sequenced and assembled eight chloroplast genomes from *I.* ser. *Batatas*. In order to have a better understanding of *Ipomoea* chloroplast genomes, we also included 32 published chloroplast genomes for comparison. Our two aims were as follows: first, to understand the conservation and diversity of the *Ipomoea* chloroplast genome through comparative genomics approaches and second, to analyze the repeat sequences in the *Ipomoea* genus.

# MATERIALS AND METHODS

## Sampling and DNA extraction

Total genomic DNA was extracted from eight species for sequencing (Table 1). The seeds of *I. trifida* (CIP 460377) were provided by the International Potato Center (Lima, Peru), those of *I. triloba* (NCNSP0323) were provided by North Carolina State University, and the rest were provided by the US National Plant Germplasm System (images of three species are shown in Supplementary Figures). Plants were grown in the greenhouse of the Xuzhou Sweet Potato Research Centre in China. Young leaves were collected from one plant of each species and were subsequently frozen in liquid nitrogen and stored at −80 °C until further use. Total genomic DNA was extracted using the Takara miniBEST plant genomic DNA extraction kit (Dalian, China). The integrity of genomic DNA was assessed by performing gel electrophoresis using a 1% agarose gel.

## Chloroplast genome assembling and annotation

We constructed genomic libraries using the TruSeq DNA Nano kit with a DNA insert size of 350 bp. Sequencing was conducted on the Illumina X Ten platform, which

generated at least 21 Gb raw data from each species. Sequence data were quality trimmed using SOAPnuke (with the options: -n 0.1 -| 20 –q 0.1 -5 1) (*Chen et al., 2018*). Sequences were assembled into contigs according to the protocol described by *Hahn, Bachmann & Chevreux (2013)* using MIRA sequence assembler software (default settings) with the reference genome *I. trifida* (accession number: KF242476.1). The reads were blasted into contigs with perl scripts and contigs were extended to obtain the complete sequence. Finally, we blasted the reads to the complete sequence, detected the coverage of reads and manually corrected the sequence. The services of library construction, sequencing, and assembly were provided by Macrogen (http://www.macrogencn.com/sy, Shenzhen, China).

The eight chloroplast genome sequences were initially annotated using the online CpGAVAS (*Liu et al., 2012*) software with default settings, and then manually corrected using Genious 11.0.5. The circular chloroplast genome maps were constructed using the OrganellarGenome DRAW tool (*Lohse, Drechsel & Bock, 2007*).

## Repeat structure analysis

Simple sequence repeats (SSRs) were detected using MISA-web (*Beier et al., 2017*). Thresholds of 10, six, and four repeat units for mono-, di-, and trinucleotides were used, respectively; while a threshold of three was used for tetra-, penta-, and hexanucleotides, respectively. REPuter (*Kurtz et al., 2001*) was used to visualize forward, palindrome, reverse, and complementary sequences, with a minimum repeat size of 30 bp and a sequence identity >90%.

## Divergence hotspot identification

In order to study the differences among these genomes, the chloroplast genome sequences were aligned using MAFFT v7.307 (*Katoh & Toh, 2010*) including whole chloroplast genomes, then sliding window analysis was conducted. To calculate the proportion of mutational events, we used a modified formula previously described by *Gielly & Taberlet (1994)*: proportion of mutational events = $[(NS + ID)/L] \times 100\%$, where NS, number of nucleotide substitutions; ID, number of indels; L, length of sequence. The step size was set to 400 bp, with an 800 bp window (*Fu et al., 2017*).

## Phylogenetic analysis

We downloaded 32 published *Ipomoea* chloroplast genomes, and *Merremia quinquefolia* (KF242501) and *Operculina macrocarpa* (KF242502) were included in the analysis as the outgroup taxa to perform the phylogenetic analyses (Table S1). Before maximum-likelihood (ML) analyses, 42 chloroplast genomes were aligned using the MAFFT plugin in Geneious 11.0.5. The gaps in the alignment were stripped. ML analyses were performed using RAxML-HPC2 on XSEDE with 1,000 bootstrap replicates and the GTRGAMMA model on CPIRES Science Gateway (*Miller, Pfeiffer & Schwartz, 2010*) (https://www.phylo.org/).

**Table 2 The chloroplast genome features of eight newly sequenced *Ipomoea* species.**

| | *Ipomoea trifida* | *Ipomoea cynanchifolia* | *Ipomoea splendor-sylvae* | *Ipomoea cordatotriloba* | *Ipomoea lacunosa* | *Ipomoea* X *leucantha* | *Ipomoea tabascana* | *Ipomoea triloba* |
|---|---|---|---|---|---|---|---|---|
| Total cpDNA size | 161,531 | 161,386 | 161,721 | 161,242 | 161,446 | 161,296 | 161,225 | 161,269 |
| Length of LSC region | 87,606 | 87,591 | 88,004 | 87,581 | 87,575 | 87,598 | 87,583 | 87,625 |
| Proportion of LSC (%) | 54.23 | 54.27 | 54.42 | 54.32 | 54.24 | 54.31 | 54.32 | 54.33 |
| Length of IR region | 30,940 | 30,882 | 30,833 | 30,806 | 30,910 | 30,840 | 30,798 | 30,810 |
| Proportion of IRs (%) | 38.31 | 38.27 | 38.13 | 38.21 | 38.29 | 38.24 | 38.20 | 38.21 |
| Length of SSC region | 12,045 | 12,031 | 12,051 | 12,049 | 12,051 | 12,018 | 12,046 | 12,024 |
| Proportion of SSC (%) | 7.46 | 7.45 | 7.45 | 7.47 | 7.46 | 7.45 | 7.47 | 7.46 |
| Total GC content (%) | 37.52 | 37.53 | 37.52 | 37.55 | 37.53 | 37.55 | 37.55 | 37.54 |
| LSC | 36.1 | 36.1 | 36.1 | 36.2 | 36.2 | 36.1 | 36.1 | 36.1 |
| IR | 40.48 | 40.52 | 40.58 | 40.57 | 40.5 | 40.57 | 40.58 | 40.58 |
| SSC | 32.29 | 32.32 | 32.23 | 32.29 | 32.28 | 32.3 | 32.26 | 32.24 |
| Total number of genes | 121 | 121 | 121 | 121 | 121 | 121 | 121 | 121 |
| Protein encoding | 80 | 80 | 80 | 80 | 80 | 80 | 80 | 80 |
| tRNA | 37 | 37 | 37 | 37 | 37 | 37 | 37 | 37 |
| rRNA | 4 | 4 | 4 | 4 | 4 | 4 | 4 | 4 |

# RESULTS

## Genome sequencing and assembly

At least 21 Gb of raw data from each species were generated using Illumina sequencing technology, then through filtering, we obtained 12.58–17.77 Gb of clean data. Using the *I. trifida* (accession number: KF242476.1) genome as a reference, we obtained the mapped reads. Finally, there were 0.59–1.28 Gb remaining bases and we assembled the chloroplast genomes of eight *Ipomoea* species using these data. The coverage of chloroplast genomes in each species ranged from 3,664 (in *I. splendor-sylvae*) to 7,941 (in *I. cordatotriloba*) (Table S2). All eight newly-sequenced chloroplast genomes were submitted to GenBank (accession numbers: MH173253; MH173254; MH173257; MH173259–MH173263) (Table 1).

## Genome features

### Genome size and GC content

The chloroplast genome size of the eight *Ipomoea* species ranged from 161,225 (*I. tabascana*) to 161,721 bp (*I. splendor-sylvae*) (Table 2; Fig. 1). These genomes displayed the typical circular quadripartite structure, consisting of a pair of inverted repeat (IR) regions (30,798–30,940 bp) separated by a large single copy (LSC) region (87,575–88,004 bp) and a small single copy (SSC) region (12,018–12,051 bp). The average guanine-cytosine content is approximately 40.5% in the IR regions, 36.1% in the LSC region, 32.2% in the SSC region, and 37.5% in the entire sequence of all plastomes (Table 2).

### Genes

The chloroplast genomes of these wild species included 80 protein-coding genes, four rRNAs (*rrn*23, *rrn*16, *rrn*5, and *rrn*4.5), and 37 tRNAs. Based on their predicted

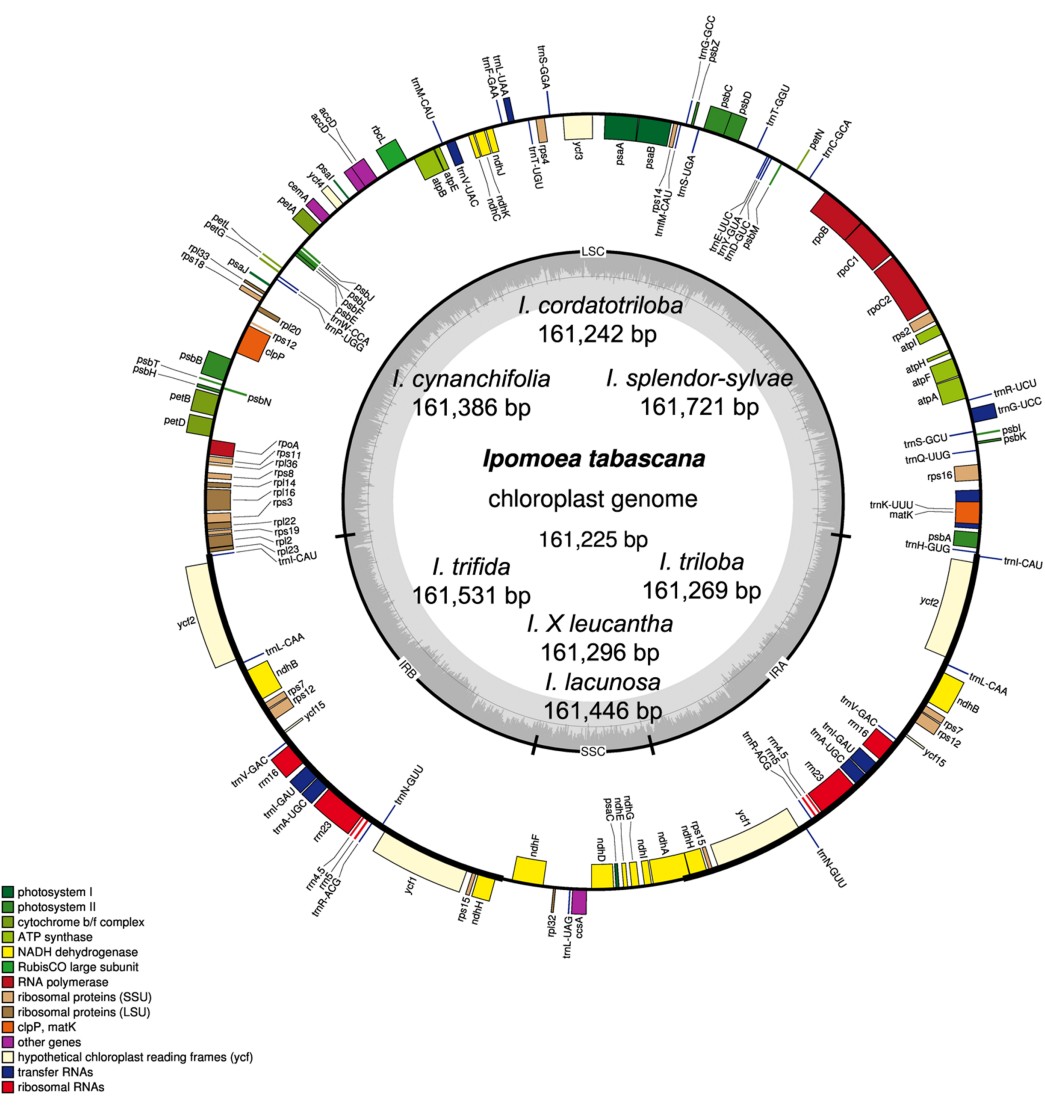

**Figure 1 Chloroplast genome map of eight *Ipomoea* species.** The inside genes of the outer circle are transcribed counterclockwise while the genes outside are transcribed clockwise. The dashed area in the inner circle indicates the GC content. LSC, large single copy; SSC, short single copy; IR, inverted repeats.

functions, these genes can be divided into four categories, (1) genes related to photosynthesis; (2) genes related to self-replication; (3) genes related to the biosynthesis of cytochrome, protein, etc., and (4) functionally unknown *ycf* genes (Table S3). A total of 73 single-copy genes located in the LSC/SSC regions and seven two-copy genes in the IRs. In the chloroplast genomes analyzed, there were 16 genes harboring introns. In these genes, 14 genes had only one intron; *ycf3* and *clpP* had two introns each (Table S4).

### Codon usage

The chloroplast genomes of the eight *Ipomoea* species we studied contained 23,766–23,804 codons in total, possessing similar codon usage distribution. AUU (Ile) was the most

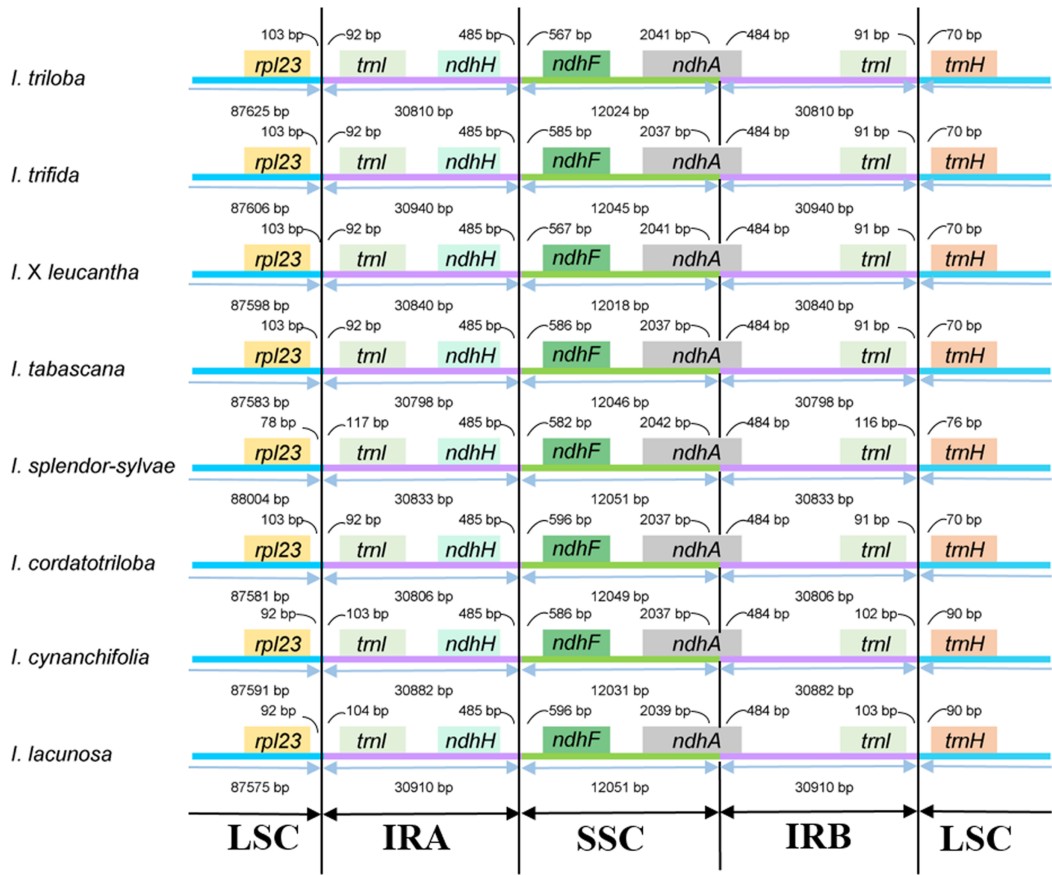

**Figure 2 Comparison of the boundaries between LSC/SSC and IR regions among the eight *Ipomoea* chloroplast genomes.**

abundant codon in all samples (Table S5). The relative synonymous codon usage of the third position showed more A/T than C/G.

### Boundary between LSC/SSC and IRs

The junctions of LSC/IRa, SSC/IRa, and LSC/IRb regions are located in the IGS region between *rpl23* and *trnI*; *ndhH* and *ndhF*; and *trnI* and *trnH*, respectively, and the location of the SSC/IRb junction within the coding region of the *ndhA* gene made pseudogenes of the *ndhA* gene of 484 bp in length (Fig. 2). The distance from *ndhH* to the IRA/SSC boundary and the length of the *ndhA* gene located in the IRB is the same.

### Repetitive sequence analysis

In total, we used 40 *Ipomoea* species for comparison (Table S1). The number of repetitive sequences in these chloroplast genomes ranged from 57 (*I. pedicellaris*) to 89 pairs (Table S6). Half of the chloroplast genomes had 88 or 89 pairs of repeats. The forward and palindromic repeats occupied 50.37% and 49.63% of the total repeats, respectively (Figs. 3A and 3C). The number of forward repeats ranged from 19 (*I. pes-caprae*) to 50 (*I. biflora*) and the number of palindromic repeats ranged from 28 (*I. pedicellaris*) to 61 (*I. pes-caprae*) (Table S6). The most common repetitive sequence lengths were 30–59 bp
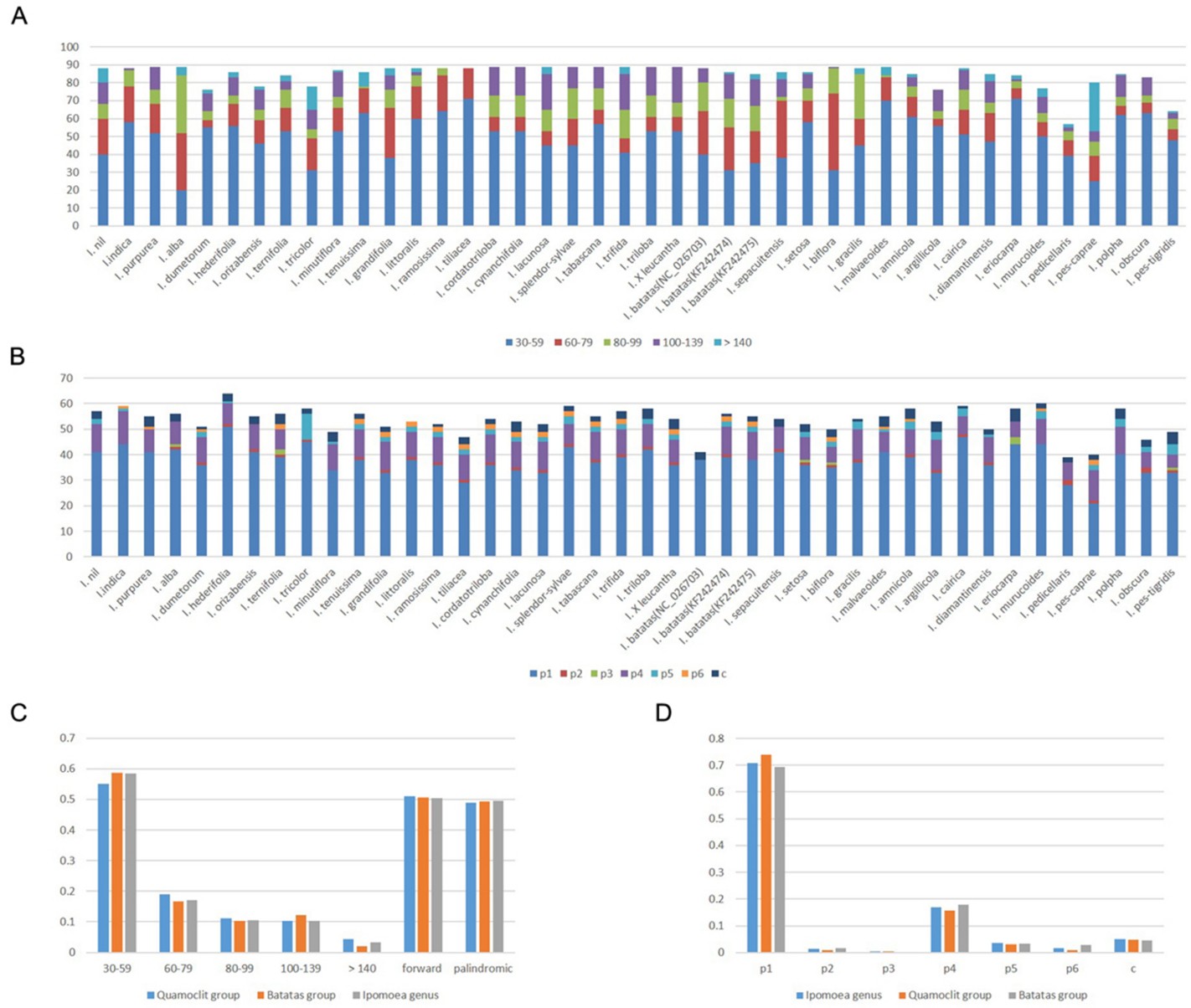

**Figure 3 Repeats in 40 *Ipomoea* chloroplast genomes.** (A) Number of repetitive sequences by lengths; (B) Number of different types of SSRs (p1–p6 indicate mono-, di-, tri-, tetra-, penta-, hexanucleotides, respectively. c indicates complex.); (C) Summary of repetitive sequences by lengths and types in *Ipomoea* genus, Quamoclit group, and Batatas group; (D) Summary of SSRs in *Ipomoea* genus, Quamoclit group, and Batatas group.

(58.55%), while those with >140 bp were least abundant (3.40%) (Figs. 3A and 3C). Most of the repeats were located in the *ycf1* gene region (most abundant region), *ycf2* gene region, and the intergenic regions between *trnN-GUU* and *ycf1* or *trnI-CAU* and *ycf2*; all of which were in IR regions. Those repeats distributed in the LSC region were mostly in the *accD* gene, *ycf3* gene, and the intergenic regions between *rpl23* and *trnI-CAU*. We also identified few repeats located in *ndhH-ndhF* within the SSC regions (Table S6). The separate analysis of the two large clades (Batatas clade and Quamoclit clade) showed similar results with genus *Ipomoea* (Fig. 3C). Interestingly, the *I. nil* belonging to the

Quamoclit clade was different from the other species. It had 43.2% (38 in 88) repeats distributed in the intergenic regions between *ycf2* and *trnI-CAU* (Table S6).

Simple sequence repeats are tandemly-repeated nucleotides in DNA sequences. We found the *Ipomoea* chloroplast genomes we analyzed contained 39–64 SSRs. The most common SSRs were mononucleotides which accounted for 70.77%, followed by tetranucleotides which accounted for 16.95% (Figs. 3B and 3D). Almost all of the mononucleotide repeat sequences were comprised of A/T repeats, while the tetranucleotides varied among different species. All the dinucleotide repeats were AT/TA repeats comprising 1.51% of the total SSRs. The proportion of pentanucleotides and hexanucleotides were 3.56% and 1.78%, respectively, while not all the species had them. Trinucleotides accounted for 0.40% in all the SSRs and were the least abundant, with only 10 trinucleotides detected across all the 40 *Ipomoea* species (Figs. 3B and 3D). Except for the mononucleotides and dinucleotides, five tetranucleotide repeats (AATA, CAAT, GAAA, TATC, and TTTC) were shared in the Quamoclit clade (Table S7). In the Batatas clade, more tetranucleotides were shared including AAAT, AATA, AGAT, ATAG, CAAT, GAAA, TATC, and TTTC. The tetranucleotide GAAA was only absent in *I. splendor-sylvae* (Table S7). In addition, the pentanucleotides (TTCTA) were present in 93.7% (15/16) and the hexanucleotides were present in 81.2% (13/16) of the Batatas clade. SSRs are different from the repetitive sequences identified by REPuter; they are almost all located in LSC regions (Table S7).

## Divergence hotspot regions

We conducted divergence analysis of the *Ipomoea* genus, including 40 species. Additionally, separate analysis in the Batatas group (16 species) and Quamoclit group (10 species) were conducted. Across the *Ipomoea* genus, the percentage of identical sites was 78.5%. The mean value of the variation was 5.12% and six regions which had a variation rate >13% were thought to be highly variable regions. Five of them were intergenic regions (*ndhA* intron, *ndhH-ndhF*, *ndhF-rpl32*, *rpl32-trnL*, *rps16-trnQ*) and one was in a gene coding region (*ndhF*) (Fig. 4A). Except *rps16-trnQ* which was located in the LSC region, all other highly variable regions were located in the SSC region.

In the Batatas group, the percentage of identical sites was 94.8% and the mean value of the variation was 1.26%. The regions with a variation rate >3.2% were considered to be highly variable regions. They were *trnG*, *rpl32-trnL*, *ndhA* intron, *ndhF-rpl32*, *ndhH-ndhF*, *ccsA-ndhD*, *trnG-trnR*, and *pasA-ycf3* (Fig. 4B). One of these was located in the IRA/SSC boundary (*ndhH-ndhF*), three of eight were located in the LSC region, and four of them were located in the SSC region (*rpl32-trnL*, *ndhA* intron, *ndhF-rpl32*, and *ccsA-ndhD*).

In the Quamoclit group, 10 species showed highly consistent sequences, with 92.0% identical sites. The mean value of variation was 2.66%. Eight relatively high variable regions with a variation rate >7.5% were detected, including *ndhA* intron, *petN-psbM*, *rpl32-trnL*, *trnG-trnR*, *trnK-rps16*, *ndhC-trnV*, *rps16-trnQ*, and *trnG* (Fig. 4C). Two of them were in the SSC region (*rpl32-trnL*, *ndhA* intron), the others were located in the LSC region, and all of them were from non-coding regions.

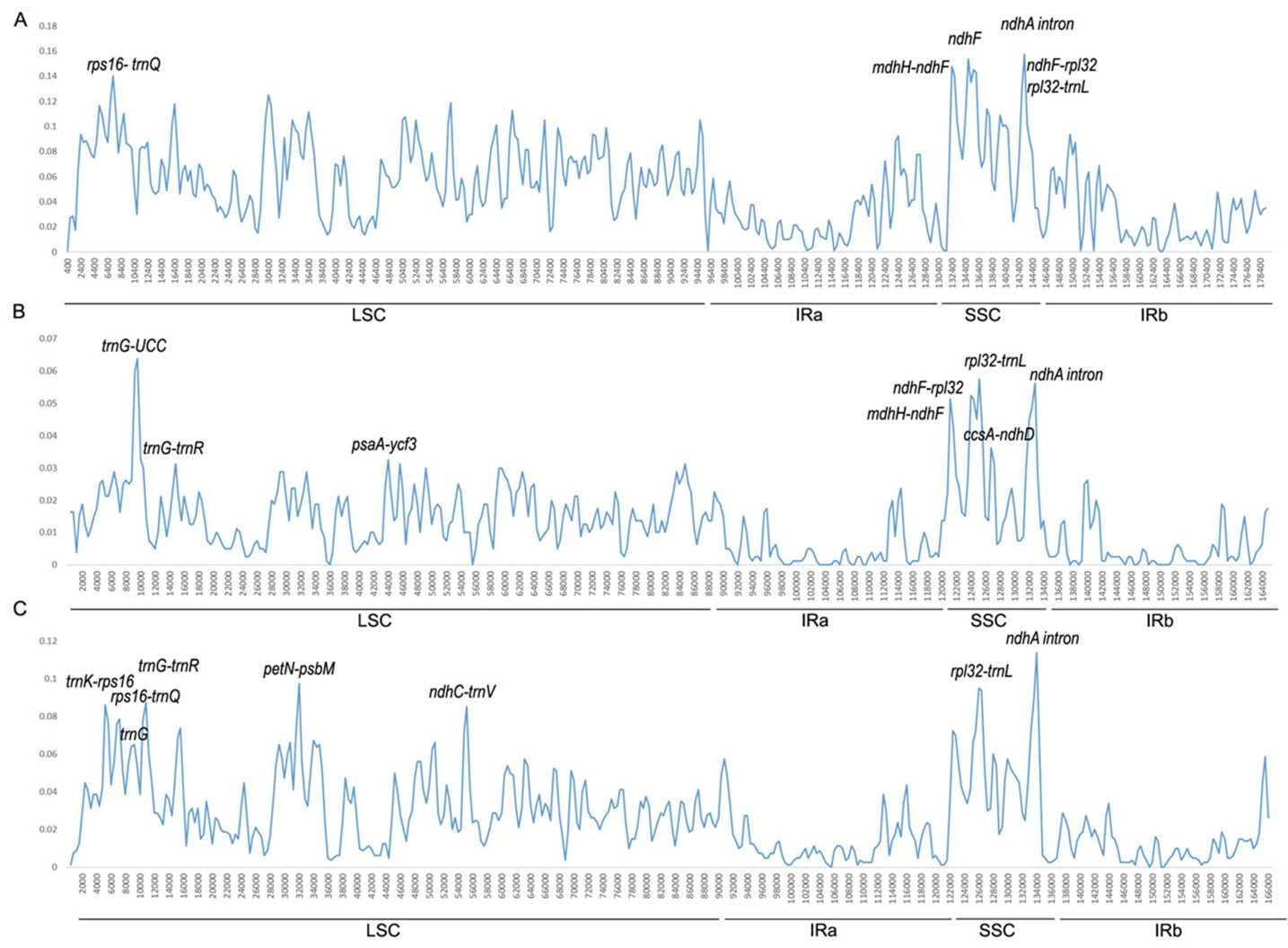

**Figure 4 Percentages of variable sites in homologous regions among the *Ipomoea* chloroplast genomes.** (A) In 40 *Ipomoea* species; (B) Batatas group; (C) Quamoclit group.

## Phylogenetic analysis

The phylogenetic tree was constructed based on 40 *Ipomoea* whole chloroplast genomes, using *M. quinquefolia* (KF242501) and *O. macrocarpa* (KF242502) as outgroups (Fig. 5). These 40 *Ipomoea* species were divided into two major clades and were further divided into seven small clades with strong support values, including Batatas, Murucoides, Pes-caprae, Quamoclit, Cairica, Obscura, and Pes-tigridis (*Eserman et al., 2014*). The Cairica, Pes-tigridis, and Obscura groups formed one major clade and the other four groups formed another clade. The support value for the Cairica clade was lower than those of other small clades. Among these seven groups, two larger groups, Quamoclit and Batatas, included the most species. *Ipomoea setosa* and *I. sepacuitensis* were located as the sister group of *I.* ser. *Batatas* and constituted the Batatas group. In *I.* ser. *Batatas*, *I. splendor-sylvae* was located at the most basal position. In the Quamoclit group, *I. dumetorum* was clustered as the sister group with others.

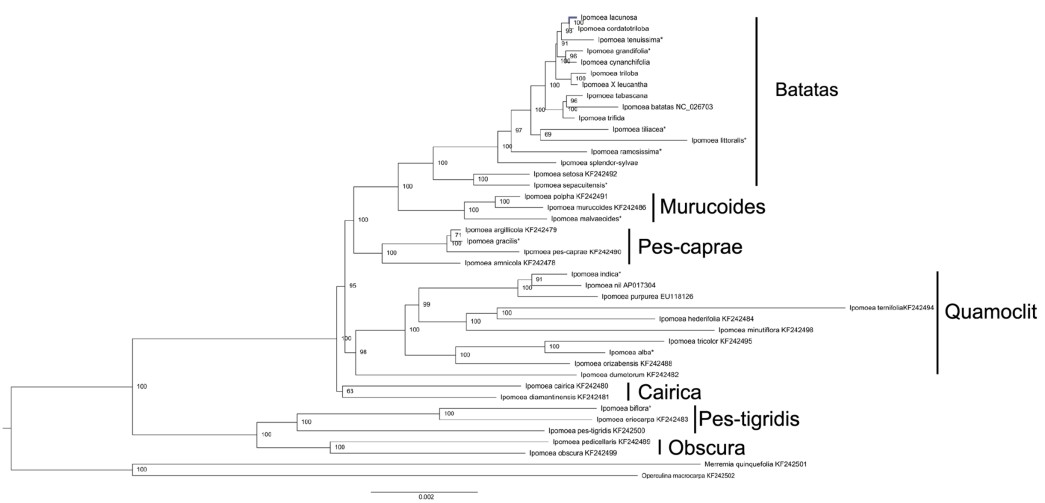

**Figure 5 Phylogenetic tree reconstruction of 40 *Ipomoea* species based on whole chloroplast genomes using *Merremia quinquefolia* and *Operculina macrocarpa* as outgroups.** *indicate the species from *Muñoz-Rodríguez et al. (2018)*.

## DISCUSSION

### Variations among the *Ipomoea* species

In the present study, eight chloroplast genomes in *I.* ser. *Batatas* were assembled. They were highly conserved in terms of genome structure and size. They displayed a typical quadripartite structure and length of the chloroplast genome sequence in the eight-wild species, ranging from 161,225 to 161,721 bp. The expansion or contraction of IR regions was thought to be the main reason for the divergences in chloroplast genome size (*Ravi et al., 2008*). Two models were proposed to explain the expansion of the IR region. Gene conversion is used to explain the small IR expansion and movements, and double-strand DNA breaks and recombination were considered to be the reason for major IR expansion (*Goulding et al., 1996*). Among the eight species, the LSC/IRa/SSC/IRb boundary genes were highly conserved, with slight structural variations and there was no significant expansion/contraction of IRs among these species. According to the studies conducted by *Eserman et al. (2014)*, the SC/IR boundary was highly conserved even across the whole tribe Ipomoeeae with some exceptions only occurring in a few species. Our eight chloroplast genomes contained 121 genes and the *infA* gene, which encodes a translation initiation factor and is almost lost in all rosid species, has also been lost in our species (*Millen et al., 2001*; *Gitzendanner et al., 2018*).

Repeat motifs are thought to have a significant impact on genome phylogeny and rearrangement (*Yue et al., 2008*). Larger and more complex repeat sequences may play an important role in the rearrangement of chloroplast genomes and sequence divergence (*Timme et al., 2007*; *Weng et al., 2014*). Here, we conducted repeat analysis in 40 *Ipomoea* species and separate analyses in the Batatas and Quamoclit groups. In *Ipomoea*, only forward and palindromic repeats were found, and they were almost equal in measure in each species. The repeats distributed mainly in the *ycf1* and *ycf2* genes. Interestingly, *I. pes-caprae* had 19 forward repeats and 61 palindromic repeats; in *I. nil*, 43.2% of

repeats were located in the *ycf2 -trnI-CAU* region rather than in the *ycf1* and *ycf2* genes. Additionally, in contrast with the Quamoclit group, the Batatas group had a slightly higher proportion of 30–59 bp repeats and a lower proportion of 60–79 bp and >140 bp repeats.

We also investigated and compared the numbers and distributions of SSRs across the 40 *Ipomoea* species. Compared with the long repeats, the SSRs were distributed more widely throughout the chloroplast genomes and were usually located in LSC regions. Most of the SSRs were found in non-coding regions; only a few were located in the coding regions (e.g., *rpoC2*, *rpoB*, *atpB*, *ycf1*, *ycf2*, and *ndhF*). The predominant type of SSRs were mononucleotides and almost all of them were A or T repeats. This is consistent with the previous findings that suggested that chloroplast SSRs are generally comprised of short polyA or polyT repeats and rarely contain tandem G or C repeats (*Kuang et al., 2011*). By comparing the Quamoclit and Batatas groups, extremely rare trinucleotides were detected in the Batatas group (0.1%) and rare hexanucleotides were detected in the Quamoclit group (0.89%). Because of the high polymorphism of SSRs in the chloroplast genome, SSRs are potentially important molecular markers in the analysis of plant population genetics as well as evolutionary and ecological studies (*Kuang et al., 2011*). In sweet potato, chloroplast SSRs ($(A)_n$ and $(T)_n$) combined with nuclear SSRs have been used to investigate the genetic diversity of sweet potato and further provided strong support for the prehistoric transfer of sweet potato (*Roullier et al., 2011*, *2013a*).

In addition, nucleotide substitution (SNVs, indels, and proportions of variability) may play a critical role in plant evolutionary processes. We conducted three independent analyses which included one in the *Ipomoea* genus and two in small clades (Batatas and Quamoclit groups) and they showed different divergence regions. Only two regions presented in all the three analyses—*rpl32-trnL* and *ndhA* intron. One of them (*rpl32 -trnL*) has been used in previous studies to disentangle the origins of sweet potato (*Roullier et al., 2013b*). Like other angiosperms (*Liu et al., 2017*), the IR regions were more conserved than the SC regions in *Ipomoea* chloroplast genomes. This phenomenon possibly occurred because of copy correction between IR sequences by gene conversion (*Khakhlova & Bock, 2006*).

## Phylogenetic relationships

The phylogenetic relationships in tribe Ipomoeeae have been constructed based on whole chloroplast genomes of 30 species including *Ipomoea* and nine other genera (*Eserman et al., 2014*). Here, we constructed a phylogenetic tree of the *Ipomoea* with 21 chloroplast genomes downloaded from GenBank, eleven from Muñoz-Rodríguez's group's research (*Muñoz-Rodríguez et al., 2018*), and our eight chloroplast genomes. Our results also divided the 40 *Ipomoea* species into seven groups that confirmed previous studies (*Eserman et al., 2014*). Determination of taxonomy and species in *I.* ser. *Batatas* is particularly difficult because individuals often exhibit intermediate morphologies between descriptions of named species (*McDonald & Austin, 1990*; *Austin, 1978*), and several species may be of hybrid origin (*Diaz, Schmiediche & Austin, 1996*). Phylogenetic analysis in *I.* ser. *Batatas* has been performed using DNA markers, such as RFLP,

RAPD, ISSR, chloroplast restriction site variation, gene sequences, and morphological analyses (*Rajapakse et al., 2004*; *Huang & Sun, 2000*; *Jarret & Austin, 1994*; *Jarret, Gawel & Whittemore, 1992*). These studies have indicated the phylogenetic relationships between sweet potato and its wild relatives; however, the support values were low. The latest study conducted by *Muñoz-Rodríguez et al. (2018)* presented strong support for the relationships in this series based on the amounts of chloroplast genomes and nuclear data. Our studies showed a similar result as that shown in a study by *Muñoz-Rodríguez et al. (2018)*, which suggested that the chloroplast genome is a very useful tool for resolving the phylogenetic relationships of *I.* ser. *Batatas. Ipomoea setosa* and *I. sepacuitensis* were clustered, and together with *I.* ser. *Batatas* formed the Batatas group. The taxonomy of the Quamoclit clade was mostly consistent with the ITS phylogeny (*Miller, McDonald & Manos, 2004*); for example, *I. indica* was more closely related to *I. nil* than *I. purpurea*. Conversely, *I. alba* grouped with *I. tricolor* rather than with *I. nil* or *I. purpurea*.

## CONCLUSIONS

In this study, we sequenced, assembled, and annotated eight chloroplast genomes derived from close wild relatives of *I. batatas*. Based on these data, we conducted a broad range analysis in the genus *Ipomoea*. Our results showed that the chloroplast genome of these eight species is highly consistent in sequence and structure. Along with 32 published *Ipomoea* species, a detailed repeat analysis was conducted. We also identified six highly variable regions which could be useful in investigating the population genetics and biogeography of closely-related *Ipomoea* species. In addition, different divergence regions were also identified in the Batatas and Quamoclit groups.

### Funding
This work was supported by the National Natural Science Foundation of China (31771367), the Priority Academic Program Development of Jiangsu Higher Education Institutions (PAPD), the China Agriculture Research System (Grant No. CARS-10-B03), and the Colleges and Universities in Jiangsu Province plans to graduate research and innovation (KYLX16_1318). The funders had no role in study design, data collection and analysis, decision to publish, or preparation of the manuscript.

### Grant Disclosures
The following grant information was disclosed by the authors:
National Natural Science Foundation of China: 31771367.
The Priority Academic Program Development of Jiangsu Higher Education Institutions (PAPD), China Agriculture Research System: CARS-10-B03.
The Colleges and Universities in Jiangsu Province plans to graduate research and innovation: KYLX16_1318.

### Competing Interests
The authors declare that they have no competing interests.

## Author Contributions

- Jianying Sun conceived and designed the experiments, performed the experiments, analyzed the data, prepared figures and/or tables, authored or reviewed drafts of the paper, approved the final draft.
- Xiaofeng Dong performed the experiments, analyzed the data, prepared figures and/or tables, authored or reviewed drafts of the paper, approved the final draft.
- Qinghe Cao contributed reagents/materials/analysis tools, approved the final draft.
- Tao Xu performed the experiments, approved the final draft.
- Mingku Zhu contributed reagents/materials/analysis tools, approved the final draft.
- Jian Sun contributed reagents/materials/analysis tools, approved the final draft.
- Tingting Dong contributed reagents/materials/analysis tools, approved the final draft.
- Daifu Ma contributed reagents/materials/analysis tools, approved the final draft.
- Yonghua Han conceived and designed the experiments, approved the final draft.
- Zongyun Li conceived and designed the experiments, authored or reviewed drafts of the paper, approved the final draft.

## DNA Deposition

The following information was supplied regarding the deposition of DNA sequences:

The chloroplast genome sequences of ten species are accessible via GenBank accession numbers MH173253, MH173254, MH173257, and MH173259–MH173263.

## Data Availability

The data can be found at GenBank (accession numbers MH173253, MH173254, MH173257, and MH173259–MH173263) and the Oxford Research Archive (DOI 10.5287/bodleian:yrYKneXED).

## Supplemental Information

Supplemental information for this article can be found online at http://dx.doi.org/10.7717/peerj.6563#supplemental-information.

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
