# Peer review of "A systematic comparison of eight new plastome sequences from Ipomoea L"

_PeerJ, doi:10.7717/peerj.6563_

## Round 0.1 · original submission · Major Revisions

There is no record of which herbarium holds your voucher specimens which means there is no quality control on your data source. This is a fundamental issue as researchers can not verify your plant identifications from the information provided. I would advise adding a column in Table 1 recording where the specimens are deposited and would further advise that you include images of those specimens as supplemental material.

·

Basic reporting

The general manuscript is written in clear English, the background information and cited references are mostly sufficient. Article structure is professional but some elements lack sufficient detail to be fully understood by the reader. In a few cases, the article relies on authors looking up other critical information - that information should be given in this manuscript. Below are some specific comments:

Abstract "SC/IR" should be "SSC/IR"?
line 47 insert "of" between "understanding" and "genetic"
line 73 - change to "The length of land plant chloroplast genomes..."
line 75 - insert "mostly" before "maternal"
sentence from lines 78-79 is awkward and unclear
line 100. for fix "1 %" as 1% - here and elsewhere?
line 104-105 - insufficiently clear explanation of data cleaning. Method could not be repeated.
line 106 - the paper is about genome assemblies, please give the methods, not just a citation to the method.
line 117 - the methods describe 6 repeat types, but the figure only has 5?
line 139 - how do we use a chloroplast reference to remove nuclear data? I think the authors mean they pulled matching cpDNA data from the set of all reads they didn't remove nuclear data, rather targeted matching cpDNA reads. If they only "removed nuclear data" then all the mitochondrial data would still be in the mix.
line 149 space fix "to 161,721" so the appropriate space is included
line 162 fix the spelling on cytochrome
line 246 - replace for with four
line 250-258 - this section doesn't fit the results into the context of what is already known vs novel findings. I think most of the patterns described here are well know for land plant cpDNA, so perhaps citing some relevant literature and discussing that briefly would be helpful.
line 270 and several other spots, a citation is given without the associated year.
lines 282 - 291. I'm not convinced these are solid conclusions given the study design and findings. For example, I. trifida comes out in two distinct clades on tree, so how can we conclude that it is sister to I. batatas when that is only true in one portion of the tree? We need to know more about the taxonomy of the group. Where would the type specimen of I trifida come out on this tree? Without that, the conclusion isn't meaningful.
Line 285. In phylogenetics, it is not common to say the one species is more "similar" to another based on the inferred relationships. Better so say it is more closely related. Similarity can have a very different meaning

Legend Figure 1 - What is the graph in the center of the diagram? Please describe this in the legend

Legend Figure 3. What are the five repeat types - describe in the legend (not the M&M described six repeat classes, why are there only 5 here? What are the SSR types?

Legend Figure 4. I presume a window size must have been defined to calculate a percentage? Should that be mentioned in the legend? What tool was used?

Legend Table 1. Describe what the accession # is? Where would one go to find the information on the origin of this sample?

Experimental design

This is original primary research and within the aims and scope of the journal
Questions were defined and the data generated will be valuable for others
Much of the methods have not been described in sufficient details to facilitate replication.

Validity of the findings

Some of the conclusions are not clearly stated or supported (see comments about about the relationship between I batatas and I trifida

·

Basic reporting

I have no concerns regarding the methodology used by the authors and the descriptions provided in this manuscript. The English writing needs minor corrections and some better explanations in various sections would be useful (see my comments on the Word document). References missing include the important work done by Roullier and collaborators in several papers between 2011 and 2013 (2011, Molecular Ecology; and 2013, PNAS and PLoS ONE).

Experimental design

The experimental design is correct and the analysis can be replicated. However, it is not clear to me what is the research question that the authors addressed nor how the results presented here contribute to filling that gap. See my comments on the "Validity of the findings" section.

My main concern regarding the Experimental design is related to the identification of the specimens included in this study. Species identification in the sweet potato group is difficult, because it relies mostly on the shape of the sepals which is notably variable. Therefore, an important question would be, how did the authors identify or confirm the identity of the materials in their study? Comparing their results to those presented by Roullier et al. and Muñoz-Rodriguez et al. (2018, Current Biology), several specimens in this study are most likely misidentified. For example, the Ipomoea ramosissima specimen (PI 540711) is likely to be Ipomoea cynanchifolia, which is morphologically identical except in the shape of the fruit when fully developed (see for example Wood et al.’s 2015 study of Ipomoea in Bolivia). The fact that this I. ramosissima and the I. tiliacea specimens are not sister to each other, as shown in previous studies, raise concerns in this reviewer about the identification of the specimens. Also, the Ipomoea trifida specimen KF242476 (obtained by the authors from the data provided by Eserman et al.) is likely to be a misidentified specimen belonging to other species, rather than proof of the non-monophyly of I. trifida as hypothesised by the authors. In addition to any conclusions drawn on this specimen, extra caution must be taken when using it as reference for assembly, as its true identity is not 100% clear.

Also, from Table 1, it is not clear the provenance of the Ipomoea splendor-sylvae specimen. Does it come from Peru or from Mexico?

Validity of the findings

This manuscript presents a study of the chloroplast genomes of ten species of Ipomoea closely related to the sweet potato. As noted by the authors, two studies of whole chloroplast genomes have been published recently: Eserman et al. (2014, American Journal of Botany) and Muñoz-Rodríguez et al. Eserman et al. provided a detailed study of chloroplast genomes in Ipomoea, including three species in the sweet potato group, and Muñoz-Rodriguez et al. studied some 170 chloroplast genomes and hundreds of nuclear DNA regions of all species in the sweet potato group specifically with comparisons between species, multiple specimens per species, phylogenies of the group, etc. Muñoz-Rodriguez et al. built on the work done previously by Roullier and collaborators and several other groups in the late 1990s and early 2000s using DNA barcodes and other markers. All those studies and especially Muñoz-Rodriguez et al. helped clarify, to a great extent, the relationships between the species in the sweet potato group. Furthermore, those studies made available the chloroplast genomes analysed, which means that around 200 whole chloroplast genomes of species in this group are now available for phylogenetic studies.

Together with what I exposed in previous sections, a major concern related to what has been exposed in the previous paragraph is that this manuscript does not present any novelty apart from, perhaps, the detailed description of the structure of chloroplast genomes. As explained above, phylogenetic relationships between the sweet potato and its wild relatives were addressed in more extensive studies by Roullier et al and Muñoz-Rodriguez et al using nuclear and chloroplast data and multiple specimens of each species. I think this study does not provide any new insights into the question. The analyses presented here could perhaps be more relevant if the authors had included more specimens per species (which is desirable in a phylogenetic study of any group of plants these days), so their monophyly could be evaluated, but as it is now, with only one sample per species, this study does not add any new discovery to this group of species. If the authors aimed to replicate or challenge the results of previous studies, I think more sampling and combination of chloroplast and nuclear genomic data would be necessary.

Perhaps more interesting is the comparison of the chloroplast genome structure between species. However, considering 1) the extraordinary similarity between chloroplast genomes in this group (as shown by the authors), 2) that several species diverged in recent times and 3) that mutations in the chloroplast genome accumulate at slower rates than in other genomes (see for example Drouin et al. 2008 in Molecular Phylogenetics & Evolution), a highly similar or almost identical configuration is expected. This was already shown in previous studies too, although not with as much detail as presented here. Taking all this into account, I think this part of the manuscript would benefit from incorporating other species across Ipomoea and even across the family Convolvulaceae, which would cover a more extensive timeframe and different evolutionary patterns. I advise the paper could be considered for publication if such comparisons across a broader taxon sampling were incorporated.

·

Basic reporting

This manuscript reports the 10 complete plastome sequences of sweet potato and related species. It contains a substantial amount of new data. But I believe it is a simple data report paper rather than a comprehensive research paper. Several apparent errors also evident in figure and table preparation. Therefore, I recommend the through revision before publication.

Discussion on line 260-281. Discussion for the phylogenetic relationships of Ipomoea trifida is very confusing because the author used the collection numbers rather than the GenBank accession numbers in discussion. Please replace the collection numbers to the Genbank accession numbers in the text. The authors simple suggest the differences of phylogenetic relations of Ipomoea species in previous study and current species are due to taxon samplings. But the possibility of the hybridization should also address using the previous nuclear gene trees such ITS markers.

Batatas is the subcategory names under the genus Ipomoea. So, it should write to be Ipomoea ser. Batatus or Ipomoea sect. Batatus rather than Batatus name alone. Please correct the names thought the manuscript.

Fig.1. Annotation of rps12 gene is wrong. It is a divided gene. The author should present an adjust map rather than the computer generated wrong map. Please redraw the rps12 region of map.

Fig. 3A. It is not the repeat types. It looks like the repeat length.

Fig.5. Please add NCBI accession numbers after all scientific names. Please marked with bald face or different color for newly reported sequences on the tree.

Experimental design

It is well performed experiment.

Validity of the findings

The manuscript contain several clean data and new reports for 10 Ipomoea pastomes.

---

## Round 0.2 · Minor Revisions

Thank you for making the improvements to the manuscript, and especially doing the extra work to check the identification of your material. I agree with the reviewers that you could usefully do some more analysis of the data you present however the publication of the new data are useful to others even with only the moderate analysis made. You will need to improve the quality of the written scientific English and this might be the right time to use a professional editing service. With the minor changes and edits to the English this paper will be ready for publication.

·

Basic reporting

This is an improved version of the manuscript, and I acknowledge that authors have made all efforts to address the different issues and questions raised by all three reviewers. Most questions have been carefully resolved, which is much appreciated.

However, I still think there is room to improve the English writing, and some sentences need re-writing. For example, sentence in line 132 is not clear: "The step size was set with to 400 bp, with an 800 bp as a window (Fu et al., 2017)". This is only one example, but authors must note that there are several other parts of the text where inconsistencies appear, singular/plural, typos, etc. I think it is necessary to correct these before the manuscript is published. It should not take much time and would help future readers of the paper.

Experimental design

No comment.

Validity of the findings

Same as in my previous review.

·

Basic reporting

This manuscript reported eight new chloroplast genome sequences of Ipomoea. With other published 32 sequences, the authors reconstructed the phylogenetic tree of Ipomoea. I wish more detailed phylogenetic anlyses and detailed description about the phylogeny of Ipomoea ser. Batatas using whole chloroplast sequences. But, I think the manuscript contains publishable information. The revised version of manuscript is better than the previous version.

But, the current version also need minor polishing befopre the submission of final version.

-The gene number counts in Table 2 should be given by unique gene numbers rather than the duplicated countings in IR regions.

-There are many grammatical errors. Please proofread the manuscript before the submission of final version.

For examples:

Line 75. The length – The lengths
Line 90. repeats varieties - repeat sequences
Line 293. Delete genus
Lines 297, 300. Ser. or Series should be use constantrly across the manuscript.
Line 311. different – difference
Line 316. A detailed repeats – A detailed repeat.

Experimental design

Experimentally, it is a standardized NGS and annotation process.

Validity of the findings

Eight new plastome sequences from seet potato plants.

Additional comments

I wish more detailed phylogenetic anlyses and detailed description about the phylogeny of Ipomoea ser. Batatas using whole chloroplast sequences. But, I think the manuscript contains publishable information.

---

## Round 0.3 · accepted · Accept

I have read the manuscript and am happy that the amendments meet the requirements of PeerJ.

#